# Two-Component System Sensor Kinases from Asgardian Archaea May Be Witnesses to Eukaryotic Cell Evolution

**DOI:** 10.3390/molecules28135042

**Published:** 2023-06-28

**Authors:** Felipe Padilla-Vaca, Javier de la Mora, Rodolfo García-Contreras, Jorge Humberto Ramírez-Prado, Nayeli Alva-Murillo, Sofia Fonseca-Yepez, Isaac Serna-Gutiérrez, Carolina Lisette Moreno-Galván, José Manolo Montufar-Rodríguez, Marcos Vicente-Gómez, Ángeles Rangel-Serrano, Naurú Idalia Vargas-Maya, Bernardo Franco

**Affiliations:** 1Departamento de Biología, División de Ciencias Naturales y Exactas, Universidad de Guanajuato, Noria Alta s/n, Guanajuato 36050, Mexico; 2Departamento de Genética Molecular, Instituto de Fisiologia Celular, Universidad Nacional Autonoma de Mexico, Circuito Exterior s/n, Mexico City 04510, Mexico; 3Departamento de Microbiología y Parasitología, Facultad de Medicina, Universidad Nacional Autónoma de México, Mexico City 04510, Mexico; 4Unidad de Biotecnología, Centro de Investigación Científica de Yucatán, A. C., Mérida 97205, Mexico

**Keywords:** two-component systems, sensor kinase, *Asgardarchaeota*, syntrophic metabolism, eukaryotic origin

## Abstract

The signal transduction paradigm in bacteria involves two-component systems (TCSs). *Asgardarchaeota* are archaea that may have originated the current eukaryotic lifeforms. Most research on these archaea has focused on eukaryotic-like features, such as genes involved in phagocytosis, cytoskeleton structure, and vesicle trafficking. However, little attention has been given to specific prokaryotic features. Here, the sequence and predicted structural features of TCS sensor kinases analyzed from two metagenome assemblies and a genomic assembly from cultured Asgardian archaea are presented. The homology of the sensor kinases suggests the grouping *of Lokiarchaeum* closer to bacterial homologs. In contrast, one group from a *Lokiarchaeum* and a meta-genome assembly from *Candidatus Heimdallarchaeum* suggest the presence of a set of kinases separated from the typical bacterial TCS sensor kinases. AtoS and ArcB homologs were found in meta-genome assemblies along with defined domains for other well-characterized sensor kinases, suggesting the close link between these organisms and bacteria that may have resulted in the metabolic link to the establishment of symbiosis. Several kinases are predicted to be cytoplasmic; some contain several PAS domains. The data shown here suggest that TCS kinases in Asgardian bacteria are witnesses to the transition from bacteria to eukaryotic organisms.

## 1. Introduction

The discovery of the superphylum *Asgardarchaeota* has introduced the possibility of finally identifying the Last Eukaryotic Common Ancestor [1]. The key finding supporting this is the high abundance of eukaryotic proteins (identified as eukaryotic signature proteins) in the 162 complete or nearly complete genomes belonging to this superphylum [2]. This has also fueled the debate regarding the topology of the tree of life, i.e., whether it is a two- or three-branched tree, implying that all eukaryotic lifeforms originated from archaea or were derived from a branch of bacteria that gave origin to archaea, which further branched to provide rise to eukaryotic organisms [3].

Much debate is also focused on the types of Asgardian archaea that most closely resemble eukaryotic organisms, with findings suggesting that *Lokiarchaeum* and *Heimdallarchaeota* are the phyla linking bacteria and eukaryotes [3]. Careful phylogenetic analysis has indicated that the tree of life has a two-domain architecture instead of a three-domain architecture, where Asgardian archaea are the ancestors of Eukarya. This evolution may have occurred in a stepwise manner [3].

With genomic sequencing, other interesting features of Asgardian archaea can be explored, such as the link between proteins with a clear prokaryotic origin and a defined function, which can also shed light on the symbiotic relationship of Asgardian archaea and the metabolic relationship with alpha-proteobacteria, suggesting that the formation of endosymbionts led to the origination of the eukaryotic lineage. The most likely candidate proteins are two-component system (TCS) sensor kinases. TCSs link the environment and the molecular and metabolic response to the constantly changing environment. Some studies have examined multiple-signal perception [4] by TCSs and their relationship with the domain Bacteria. There are many questions regarding how Asgardian archaea established a close relationship with bacteria to give rise to eukaryotic cells, specifically regarding the detection of metabolites that showed the dependence between an Asgardian bacterium and an aerobic bacterium, which led to the evolution of the functional symbiont that ultimately became the mitochondrion.

The main objectives of this research were to identify prokaryotic features in the genome and meta-genome assemblies of Asgardian bacteria and to identify proteins that may serve as indicators of Asgardian bacteria evolution. Additionally, the study aimed to identify consistent evolutionary traits that reflect the lifestyle of Asgardian bacteria and align with the current understanding of their metabolic features. Therefore, particular attention was given to investigating the two-component systems (TCS) present in these organisms. Here, sequence analysis and structural models of two sets of TCS sensor kinases were analyzed from a meta-genome assembly belonging to *Candidatus Lokiarchaeum* sp. GC14_75 and a fully sequenced genome from a cultured Asgardian archaeon, *Candidatus Prometheoarchaeum syntrophicum* MK-D1. Our results suggest that both the meta-genome assembly and full genome assembly encode divergent sensor kinases with peculiar architectures, which are predicted not to be bound to the cell membrane and contain multiple PAS domains. Genome sequencing of *C. Prometheoarchaeum syntrophicum* MK-D1 revealed homologs of well-characterized TCS sensor kinases. Overall, the structural features analyzed here suggest the presence of novel sensing mechanisms in senor kinases in these archaea. Here, a discussion of the implications of the domains found in the sensor kinases of both genomic and meta-genome assembly sequences that may have been involved in establishing a syntropic relationship that ultimately resulted in the formation of eukaryotic cells is presented.

## 2. Results

### 2.1. Asgardian TCS Identification and Characteristics

Asgardian archaea’s genome size from the available datasets is approximately 3.3 Mb, with an average coding capacity of 3155 proteins (data obtained from Liu and colleagues [2]). The genome size is relevant to the history of the evolution of Asgardian archaea since overspecialization may have occurred due to a restrictive environment, such as the ones in which archaea thrive. For Asgardian archaea, this may also indicate a close link between the reduced number of metabolic pathways and the complementation of these pathways by neighboring bacteria, resulting in the establishment of endosymbionts that led to the origination of eukaryotic cells.

The rationale for assessing TCS sensor kinase genes is as follows. First, TCSs are important molecular signaling systems for sensing the environment and responding to ever-changing conditions and other bacteria. Thus, these systems should be specific to the Asgardian lineage or exhibit features common to other TCSs. Second, TCS diversification (if any) in the Asgardian lineage may be centered in the N-terminal region, where the signal-receiving module is usually located, and the catalytic conserved domain contains identifiable motifs and residues such as a conserved histidine, ATP-binding box, and conserved glycine resides [5,6,7,8]. Third, divergence with a restricted group of bacterial domains or strong similarity of domains/whole sequences may reinforce the idea that the Asgardian origin is deeply rooted in the domain Bacteria, as speculated [1]. The noneukaryotic features may be useful for reconstructing this aspect of the evolution of Archaea and Eukarya. Finally, the divergence (if any) with Asgardian kinases may indicate a link with eukaryotic kinases since they have been found in fungi, Mycetozoa, plants, and diatoms but not metazoans, suggesting that kinases have “permeated” the eukaryotic domain [9].

With all the above, comparing genomic datasets rendered a total of 17 kinases for *C. Lokiarchaeum* and 12 for *C. Prometheoarchaeum*, with 8 annotated sensor kinases from the *C. Heimdallarchaeota* meta-genome assembly data. Table 1 shows all the domains identified by the Conserved Domain Search. As shown in Appendix A, *C. Heimdallarchaeota* kinases contained only PAS and the catalytic histidine kinase domains. In both datasets, several domains present in the sensor kinases showed homology to known bacterial kinases, such as AdeS, AtoS, ArcB, BaeS, and TorS. TorS stands out due to its involvement in sensing trimethylamine N- oxide (TMAO) and shares homology with a BaeS catalytic domain.

Additionally, several kinases contain more than one PAS domain in their sequence. Appendix A illustrates a comparison of the TorS sensor domain with the experimentally determined TMAO-bound TorS/TorT structure from *Vibrio parahaemolyticus* [10], showing sequence and structural conservation in the kinase WP_147883838.1. The remaining kinases with TorS domains showed limited conservation with the experimentally determined structure (Appendix A). However, even though the kinases with a predicted TorS/TMAO domain are less conserved, the functionality of this domain cannot be ruled out since there are additional PAS domains present in these sensor kinases, which may be linked to sensing short-chain fatty acids or other metabolism-derived molecules.

The molecular weight of the sensor kinases ranged from 34.17 to 160.35 kDa, and the pI ranged from 4.94 to 8.89. The calculated aliphatic index indicates that although the environmental conditions should have forced the sensor kinases to be adapted to high temperatures, several kinases have a low aliphatic index (near 90). As shown in Appendix A, the kinases from *C. Prometheoarchaeum* are clustered within a certain molecular weight range and aliphatic index. In contrast, the sensor kinases from *C. Lokiarchaeum* and *C. Heimdallarchaeota* show a scattered pattern. The results correlate well with the thermophilic nature of these proteins and the origin of these organisms [11]; in addition, some kinases are within the limit for thermostable proteins in the three datasets (below 90 [12]).

### 2.2. Sequence Comparison and Analysis of Asgardian TCS

All the kinases were subjected to sequence homology analysis and phylogenetic reconstruction. In Figure 1A, a 2D pairwise map is shown, and two clusters of low homology were identified between subgroups of kinases from each dataset. Most of the Asgardian proteins showed low homology with bacterial, yeast, or plant kinases, suggesting important differences from bacterial kinases. The two kinases seem identical, even though the genome annotation indicated otherwise, but were included in all the analyses conducted.

Phylogenetic reconstruction (Figure 1B) confirmed that WP_147663673.1 and QEE16740.1 were identical (as also shown in Table 1, with the same molecular weight and pI) and that QEE16756.1 and WP_147663689.1 were 98% identical; only small differences were found related to the TorS sensing motif, but overall, the proteins had the same molecular weight and pI (Table 1). These kinases cannot be ruled out completely since they may have been the product of recent gene duplication; they could be bacterial DNA contaminants introduced during library preparation for sequencing, results of varying the depth of sequencing, sequencing artifacts, or hybrid sequences generated in the genome assembly [13,14]. Nevertheless, there are six kinases with these domains in *C. Prometheoarchaeum*, suggesting a functional role of TMAO in this lineage.

The rest of the sensor kinases showed clustering in each dataset but generated four distinctive groups with different branching points (indicated with a red arrow) (Figure 1B). The comparison was performed with all *Escherichia coli* sensor kinases, two sensor kinases from Saccharomyces cerevisiae, and two from *Arabidopsis thaliana*. Group A is the more distant group from the rest of the sensor kinases, with homology to RcsC and SLN1 kinases from *E. coli* and *S. cerevisiae*, respectively.

The phylogenetic reconstruction suggested that four groups of sensor kinases could be identified in the two datasets, where *C. Prometheoarchaeum* displayed two sets of proteins, one closer to bacterial homologs and another as an outgroup sharing homology with *C. Heimdallarchaeota* sensor kinases (Group A). Additionally, upon combining the data from the Conserved Domain Search and complementing the information with ScanProSite [15,16] (Appendix A), kinases from Group C showed the most varied domains, including more than one PAS domain.

A close inspection of the domains found in the meta-genome assembly dataset identified the kinase KKK40193.1, which contains a C-terminal domain homologous to the bacteriophytochrome light-regulated histidine kinase domain (COG4251, Table 1). This domain is found in the Conserved Domain Database associated with photosynthetic bacterial sensor kinases (UniProt Q9LCC2); this will be interesting to explore further since these organisms may experience reduced exposure to light, perhaps due to volcanic activity, and their role remains unclear.

The kinases from *C. Lokiarchaeum* are taxonomically related to homologs in the domain Archaea (Appendix A). Nevertheless, some examples from Groups B and D (Figure 1B) are taxonomically associated with the domain Bacteria, suggesting that the taxonomic and phylogenetic relationships of the sensor kinases may have had bacterial roots and then expanded to eukaryotic organisms, as shown in Group A in Figure 1B.

In Figure 1B, six of the kinases have a TorS TMAO-sensing domain; this domain is relevant for the sensing of metabolites directed to short-chain fatty acid biosynthesis and reflects a broad metabolic range for these bacteria and links the Wood–Ljungdahl pathway [17] to the primary theory of endosymbiosis, which states that organisms ultimately need to anaerobically oxidize short-chain hydrocarbons generated under hydrothermal conditions [18].

Heme-binding domains were also found in all the groups, indicating the existence of gas-sensing domains, including those for NO, CO, and even oxygen [19,20], along with the possibility of PAS domains being involved in the sensing of more complex stimuli such as light and other small molecules. The existence of heme-binding domains also suggests the relevance of oxygen in the environmental conditions of Asgardian archaea and may be linked to the forced symbiosis imposed in early oxygenic conditions.

### 2.3. Structural Models Show Domain Complexity in Asgardian TCS Sensor Kinases

Next, all kinase sequences and structural features were assessed to find relevant conservation with well-characterized protein kinases. As shown in Figure 2A, AlphaFold2 (DeepMind, London, UK) models were generated for all *C. Lokiarchaeum* and *C. Prometheoarchaeum* kinases and for *C. Heimdallarchaeota* kinases (as shown in Appendix A). The first feature that stands out is the catalytic domain (at the C-terminal end), showing a high degree of conservation among all kinases (Figure 2A,B and Appendix A). As a reference, the EnvZ kinase (PDB 4KP4) is shown in Figure 2B. The typical parallel beta-strand alpha-helix fold is shown in all kinases. The H-box (conserved histidine) and G1/G2 boxes are also conserved (shown in the WebLogo in Figure 2B). The EnvZ catalytic domain is shown as a reference with the conserved residues needed for ATP binding and the phosphotransfer reaction. Based on the reported structural features of sensor kinases, all the catalytic domains have conserved residues and exhibit the expected arrangement, as reported previously [21].

The N-terminal domains of many of the kinases showed a different fold from those of other bacterial kinases, suggesting that the association with the cell membrane was different, perhaps in a single transmembrane domain or even in the cytoplasmic region. By analysis with two various tools, ten kinases in *C. Lokiarchaeum* and *C. Prometheoarchaeum* were predicted to be associated with the cell membrane (Figure 2C). In the case of *C. Heimdallarchaeota*, only two kinases were predicted to be membrane-bound, and one (OLS17067.1) showed a predicted secretion signal peptide (Appendix A). The membrane-bound proteins in *C. Lokiarchaeum* and *C. Prometheoarchaeum* were clustered in Groups A and B, only two examples were found in Group C, and only one was found in Group D. The low abundance of evident transmembrane domains suggests that these kinases may have a role in sensing intracellular signals, which is consistent with the identification of the TorS/TMAO sensing domain and several PAS domains in these kinases. Based on the current knowledge of bacterial TCSs, this is a novel finding due to the limited number of cytoplasmic sensor kinases in the same organism.

Signal detection by sensor kinases is an important subject to address. In the sensor kinases analyzed, 18 had two or more predicted PAS domains (see also Appendix A).

A comparison of these kinases showed that most contained the conserved characteristic residues of the core domain (N34, G42, and E47, domain numbering [24]) (Figure 3A). Additionally, in many cases, PAC domains were associated with the PAS domain (Appendix A), which is proposed to be required for the PAS domain fold [4,25,26].

### 2.4. Unusual Number of PAS Domains in Asgardian TCS Sensor Kinases

The high number of sensor kinases containing a PAS domain was intriguing. Close inspection of the sequences revealed that all the PAS domains (catalytic and accessory domains listed in Supplementary File S1) showed the characteristic conserved residues of bacterial PAS domains (Figure 3A). In contrast, a comparison of the PAS domains of eukaryotic proteins showed less conserved residues than their bacterial or Asgardian counterparts. Individually modeled domains showed that the overall folding was similar among bacterial PAS domains. Still, in the case of the eukaryotic domains, the folding was different from that of the bacterial domains. The PAS domains from human proteins and those with experimentally determined structures showed an overall conserved architecture with low sequence identity. These domains were too divergent from bacterial and other PAS domains from fungi and flies (Figure 3B). Structural comparison (Figure 3C) for each group revealed that the PAS domain had a similar structure in all instances, and the same regions were conserved. Comparing two members of each category revealed that the PAS domain was highly divergent when eukaryotic counterparts were included, but the structure was highly conserved. This suggests that the PAS domains in Groups A and *C. heimdallarchaeota* are conserved and closely related to bacterial PAS domains.

Finally, phylogenetic reconstruction of the eukaryotic lineage origin has been difficult, and no clear Last Eukaryotic Common Ancestor has yet been identified. Devos [1] argues that the origin of Eukarya may be rooted in a common bacterial origin, followed by the diversification of Archaea and Eukarya [1]. In this model, bacteria of the Planctomycetes–Verrucomicrobia–Chlamydiae (PVC) group may be an intermediate between the bacterial ancestor and modern archaea and eukaryotic organisms. To test this, protein BLAST analysis of the two datasets was performed with the PVC group; the highest scoring sequences obtained were then incorporated with the sensor kinases used in this study to assess their phylogenetic position. As shown in Appendix A, incorporating PVC sensor kinases inverts the phylogeny and sets Group A further apart from the *C. Lokiarchaeum* and *C. Prometheoarchaeum* datasets (Group D). The results presented here place the Asgardian sensor kinases as divergent from bacterial kinases, suggesting that, as proposed by Devos [1], PVC bacteria serve as the link between bacterial evolution and divergence into Archaea. This positioning of Asgardian sensor kinases as witnesses of this transition reflects the importance of sensor kinases in adapting to the current conditions under which Asgardian bacteria thrive. Consequently, Asgardian bacteria ultimately played a pivotal role in the origin of eukaryotic cells through metabolic syntropy, leading to the establishment of a strong symbiotic relationship that paved the way for eukaryogenesis.

In summary, Asgardian kinases have important features that separate them from their bacterial counterparts. Nevertheless, the PAS domains stand out because they have sequence and structural features from bacterial PAS domains rather than from eukaryotic examples. The eukaryotic PAS domains showed a more extended conformation and fewer conserved residues. The more divergent PAS domains belonged to Group A, the most distant sensor kinases from the bacterial domain. They were present in the meta-genome assemblies for *C. Lokiarchaeum* and *C. Heimdallarchaeota*.

## 3. Discussion

Thus far, the hypothesis of the strong linkage of Asgardian archaea as the ancestors of eukaryotic cells has been supported by identifying proteins with eukaryotic signatures in Asgardian archaea. Nevertheless, no evidence has been provided supporting a link from the bacterial origin that suggests a directional relationship towards the evolution of eukaryotic cells.

Here, we tested two aspects by using TCS kinases. First, if the homology of these proteins indicated a bacterial origin, the direct relationship from bacteria to eukaryotic lineage would be directly explainable. If not, overspecialization and diversification in the Asgardian group of TCSs may have resulted in bacteria with horizontally transferred elements that may have a eukaryotic origin. Additionally, the type of kinases present in Asgardian archaea may further provide evidence of a metabolic relationship with other bacteria that may ultimately support the endosymbiosis between them.

The host is the first aspect to consider in the analysis of TCS sensor kinases. Asgardian archaea have relatively small genomes and a limited coding capacity, suggesting that a highly specific set of TCSs may be adapted to the environment in which these organisms thrive. Here, multiple PAS domains and mostly cytoplasmic kinases play a role in sensing metabolism-related stimuli rather than complex extracellular signals, as in other bacteria. The highly conserved catalytic domain suggested a bacterial origin and possessed specific features, such as the PAS domain. Thus, the lineage of Archaea may indeed be closely related to bacteria, as proposed by Devos [1], and may have a reduced set of TCSs for sensing specific environmental conditions, resulting in further restrictions. The analyses shown here suggest a plausible metabolic link between Asgardian archaea and bacteria producing short-chain fatty acids. An oxygen-sensing mechanism is perhaps present, resulting in “abortive” interactions with aerobic or facultative organisms, which may have contributed in the past to the establishment of true symbiosis that later gave rise to eukaryotic cells.

The finding that many sensor kinases contain several PAS domains suggests that the diversity of cytoplasmic signals these kinases can respond to is surprising. The cytoplasmic sensing repertoire for PAS domains is relevant for the theory regarding the establishment of endosymbiosis since these proteins sense pH, toluene, Zn ions, oxygen, osmolarity, stress signals (c-di-GMP), and other proteins [27]. Perhaps the implication of TCS sensor kinases having several PAS domains is based on the recent metabolic analysis of available genomes of *Asgardarchaeota* organisms, showing that the Asgardian archaea have the potential to degrade aliphatic and aromatic hydrocarbons [28] via benzoyl and malonyl coenzyme A and β-oxidation. This genetic and metabolic potential of archaea suggests a specialized ecological role.

In all instances, the regulatory network needed to detect a myriad of stimuli present in the niche where Asgardian archaea thrive suggests that the establishment of a strong symbiotic relationship is linked to the proper detection of metabolites and both environmental and intracellular conditions that ultimately result in an inseparable dependence of two (or perhaps more) organisms. Although not all PAS domains may be functional or other sensing domains may be present, they are too divergent to be detected with the approach used here. One such example is the ArcB homolog in *Haemophilus influenzae*, which lacks a PAS domain but can still sense the cell’s redox state [29].

The finding that a fully sequenced genome of an Asgardian archaeon harbored sensor kinases that were homologs to known and characterized TCS sensor kinases further supports the notion that bacteria form the root of the branches of Archaea and Eukarya lineages [1].

Another piece of the puzzle is finding a blue-light sensor kinase (KKK40193.1) in *C. Lokiarchaeum*, which is linked to the finding of putative rhodopsin and RuBisCO coding genes [30]. As proposed by Bulzu and colleagues [30], this may be a remnant of the evolutionary history of *Asgardarchaeota*, which may have thrived in light-exposed environments and on the top layer of sediments [30]. The metabolic niche for this group suggests that it is micro-oxygenic; thus, the sensor kinases analyzed here are consistent with this observation. Additionally, the *C. Prometheoarchaeum* genome reveals a more detailed and accurate signal transduction repertoire in the *Lokiarchaeum* branch, whereas, in the *Heimdallarchaeota* branch, the homology to bacterial counterparts is limited, suggesting that it is the branching point to specialized kinases and that the preservation of kinases that can sense specific metabolites and redox states was crucial for the establishment of an endosymbiotic relationship, specifically of an oxygen-utilizing endosymbiont [2]. In particular, the presence of AtoS homologs suggests the ability to sense short-chain fatty acids [31] and may be the link between the metabolic needs of Asgardian archaea and the relationship for establishing the strong metabolic crosstalk that eventually resulted in the development of endosymbiosis, e.g., acetate perception for the regulation of acetyl coenzyme A synthesis, as described previously [32]. This is consistent with the repertoire of metabolite transporters found in the *C. Prometheoarchaeum syntrophicum* MK-D1 strain and other Asgardian meta-genome assemblies previously reported that indicate that these organisms depend on the transport of all the molecules they need [33].

Interestingly, in most instances, the transporters reported by Rossum and coworkers [33] are homologous to prokaryotic transporters rather than to eukaryotic systems, which is also consistent with the findings reported here. Following the same line of thought, in *C. Prometheoarchaeum syntrophicum* MK-D1, there are 14 proteins involved in the transport of fatty anions, 15 related to lipid precursors, and 237 without inferred substrates, indicating that there is a wide range of unknown features yet to be studied as signals for AtoS homologs. Russum and coworkers’ paper (2021) [33] concluded that the *C. Lokiarchaeum* and MK-D1 strains contain twice the number of lipid precursor transporters, suggesting that the presence of two homologs to AtoS may be required for complex and perhaps non-crosstalk signal processing for metabolic adaptation. One example of this phenomenon is the sensor kinase ArcB; autophosphorylation of ArcB and transphosphorylation of its cognate response regulator ArcA are enhanced by anaerobic metabolites such as D-lactate, acetate, and pyruvate [34]. A cautionary note regarding the possibility in meta-genome assemblies is that hybrid sequences may have arisen from the assembly itself. This must be considered when analyzing multiple domains present in the same protein.

Nevertheless, in the case of the fully sequenced genome of the MK-D1 strain, the domains and features found in the present report represent further evidence that bacteria contributed to the rise of Archaea and then of the Asgard superphylum, which needed a metabolic connection to develop either true or abortive symbiosis. With the analyses shown here, the contribution of TCS sensor kinases suggests a scenario where these proteins originated in bacterial organisms and a functional expansion in bacteria. Due to restrictive environmental conditions, PVC bacteria diverged, rendering sensor kinases less conserved with their bacterial counterparts. Then, these proteins were inherited from PVC by Archaea, which ultimately diverged into Asgardian archaea, protists, fungi, and plants (Figure 4). This is further supported by the extensive genetic drivers of variation found in Asgardian archaea, such as unique mobile elements that have stabilized bacterial genes in modern-day genomes but still retain ancient signatures during the formation of eukaryotic cells [35], which is consistent with the genome size found in these organisms [35].

Further analysis of other signal transduction proteins in Asgardian archaea may lead to the identification of novel mechanisms of signal reception and processing that may result in stronger crosstalk between membrane-associated receptors and cytoplasmic kinases and even phosphorylation of non-TCS proteins, as recently proposed in fungi [36], leading to the identification a more complex regulatory network. The results presented here suggest that sensor kinases in the Asgardian bacteria are significantly different from their bacterial counterparts, warranting synthetic approaches to characterize them in detail. Additionally, these findings demonstrate extensive sequence and structural features that establish a connection between these organisms and their bacterial counterparts. Perhaps one alternative mechanism is the presence of bona fide sensor kinases and pseudo-kinases (for example, KKK40347.1, which lacks the conserved catalytic histidine), resulting in regulatory enzymes that can integrate signals into complex networks [37]. The possible mechanism underlying the prevalence of bacterial TCSs with a unique architecture in Asgardian archaea is indicated by the recent finding that integrons are present in Archaea, mostly CALINs (clusters of attCs lacking IntIs), and by the functional genes in the integrons identified [38]. Ghaly and colleagues found several active genes in integrons, which suggest that in earlier stages of eukaryotic cell development, bacterial genes played an important role in the establishment of endosymbionts (of syntrophic nature) and thus prevailed despite selective pressures imposed by the soon-to-be host [39]. Asgardian bacteria are extremely complex, and culture-based methods may allow us to reveal the role of TCSs in these organisms.

Recently, Eme and coworkers [40] published a study describing the evolution of genomes in Asgardian archaea. They observed an increased occurrence of gene duplication and a reduced frequency of gene loss. Furthermore, they noted a transition from thermophilic chemolithotrophy to a heterotrophic lifestyle, accompanied by the adaptation to mesophilic growth conditions observed in eukaryotes. This report supports the findings that the sensor kinases analyzed in this study exhibit PAS domain expansions and display a robust interaction with the current environment in which Asgardian bacteria thrive. Further investigation of the Asgardian genome sequence will provide valuable insights into the significance of both prokaryotic and eukaryotic characteristics of Asgardian bacteria. This exploration is crucial for ultimately reconstructing the evolutionary origins of these bacteria.

The results presented here are limited to the technological advance represented by AlphaFold2. Although the phylogenetic reconstruction of the available sensor kinases from Asgardian bacteria supports the observations presented, the structural features found in these sensor kinases are constrained by the current understanding of protein structure, which is based on approximately ~200,000 available protein structures in the PDB. These structures have been used to generate the training database for AlphaFold2. By using synthetic gene fragments to produce Asgardian sensor kinases in a heterologous host, the functionality of these kinases can be fully elucidated, and their structure can be experimentally assessed. Additionally, leveraging the information presented here, further investigation into additional additives in growth media may lead to faster and more robust growth conditions for Asgardian bacteria. This, in turn, could increase the number of available genomes and allow for exploring more signaling systems in Asgardian bacteria.

## 4. Materials and Methods

### 4.1. Sequence Retrieval

Protein sequences for all predicted sensor kinases were obtained from BioProject PRJNA259156 [41], for the metagenome assembly for *Candidatus Lokiarchaeum* sp. GC14_75, and BioProject PRJNA557562 [42], for *Candidatus Prometheoarchaeum syntrophicum* MK-D1, which was cultured and fully sequenced. Sequences from *Candidatus Heimdallarchaeota* were retrieved from the BioProject PRJNA319486 [43]. The analysis was restricted to annotated proteins as sensor kinases during genome annotation by text query and were present in fully sequenced contigs. Additional sequences from *Escherichia coli* were retrieved from EcoCyc [44]. *E. coli* sequences were used since the sensor kinases of this organism have been demonstrated to be active, and several studies have identified the signals that these kinases respond to. Additional sequences were obtained from the Yeast Database (https://www.yeastgenome.org/, accessed on 24 July 2022) and the *Arabidopsis* database (TAIR, https://www.arabidopsis.org/, accessed on 24 July 2022) as eukaryotic examples of sensor kinases.

### 4.2. Protein Sequence Analyses

Sequences were analyzed by Blastp [45], and conserved domain analysis was conducted in the NCBI web server [16] using the default settings to assess the conserved domains in all sensor kinases analyzed in this work. Additionally, Prosite scan [46] was used with the default settings to confirm the domains identified by Conserved Domain Search and delimit the most conserved PAS or AtoS/TorS domains to individually model them and establish structural homology between them with experimentally determined PAS domains in sensor kinases (PDB entries, indicated in each figure).

Sequence alignment was conducted with Clustal Omega [47] with the default settings, and the output was visualized as a 2D pairwise identity map by Alignment Viewer (https://alignmentviewer.org/, accessed on 24 July 2022) with the default settings. Phylogenetic analysis was conducted with MEGA version 11.0.13 [48]. The phylogenetic tree was evaluated by the maximum likelihood method with 500 bootstrap iterations. Each branch was manually colored to identify each branching point. Blastp analysis also included obtaining the Blast Tree View using fast minimum evolution and sorting by distance to determine the taxonomic group of each kinase.

### 4.3. Protein Structural Models Using AlphaFold2

Protein sequences were modeled using AlphaFold2 [49] with the default options, using the API hosted at the Söding laboratory based on the MMseqs2 server [50], generating five models per full-length protein or specific domain analyzed, using the most accurate structural alignment (pLDDT). Complementary reference structures were retrieved from the AlphaFold2 database at https://alphafold.ebi.ac.uk/, accessed on 24 July 2022, hosted by the European Bioinformatics Institute. AlphaFold2 renders good models restricted to highly dynamic regions [51]. Here, for the general purpose of assessing the structural features of sensor kinases, the models provide a good approximation of the distribution and folding of relevant signaling-related domains. Structural dynamics, such as the ones demonstrated for sensor kinases, such as the rotational on/off movement of ArcB [52], are missing from these models. Nevertheless, they provide a good approximation to the conservation of structural features.

### 4.4. Protein Structure Comparison

All protein alignments were performed with SALING using the default options [53] and analysis with UCSF-Chimaera to highlight the conserved regions in structural alignments [54]. Structural alignments indicate the SALING score for each comparison. Visualization of protein models for obtaining individual images arranged as in structural alignments was conducted with PyMOL [23]. A rainbow color scheme and cartoon representation were used to assess the N-terminal (blue) and C-terminal (red) end localization.

### 4.5. Protein Topology Analysis

Topology was evaluated with TOPCONS [22] and Protter [55] tools with the default settings. Both tools rendered the same topology for all the kinases analyzed; therefore, the Protter display was selected for presentation.

## Figures and Tables

**Figure 1 molecules-28-05042-f001:**
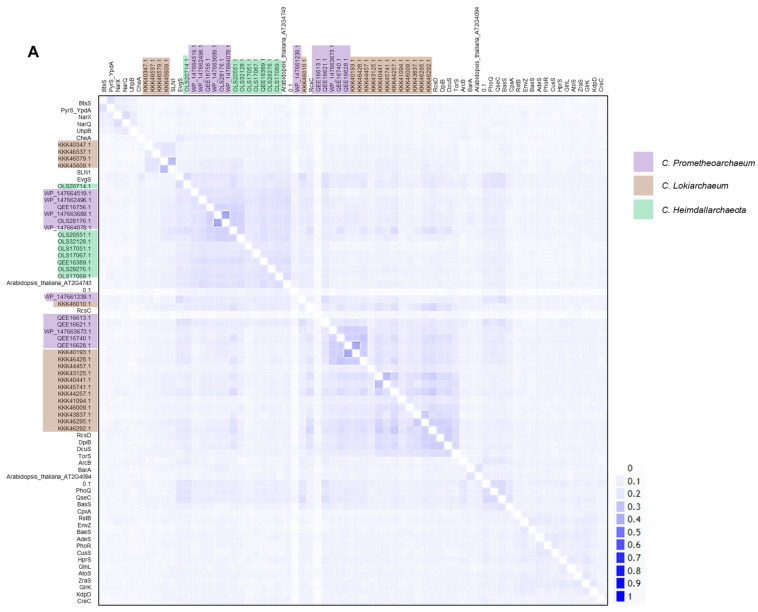
Sequence analysis reveals clusters of homologous kinases in Asgardian archaea. (**A**) A pairwise identity 2D map showing the Clustal Omega alignment of all kinases analyzed in this study. In the figure, the sets belonging to each organism are indicated by a color box (code on the **right**). Color in the 2D map indicates identity. (**B**) Maximum-likelihood phylogenetic analysis of all TCS kinases. Ancestral nodes are marked with a red arrow. Clusters derived from the latest ancestral node for Asgardian archaea are indicated. Each type of kinase is indicated by a color box (code on the **right** of the figure). Color arrows indicate special features: black (multiple PAS domains), gold (heme-binding domain), and dark blue (TorS domain).

**Figure 2 molecules-28-05042-f002:**
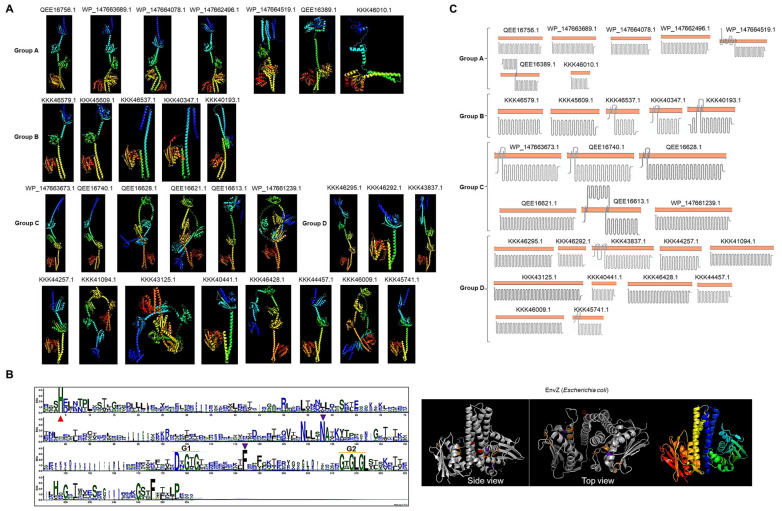
Structure and topology of *Lokiarchaeum* sensor kinases. (**A**) The AlphaFold2-predicted structures of *C. Prometheoarchaeum* and *C. Lokiarchaeum* sensor kinases are presented shown using a rainbow color scheme (blue, N-terminal end; red, C-terminal end), indicating the accession number as in Figure 1. (**B**) Clustal Omega multiple sequence alignment of all catalytic domains, represented as a WebLogo. Conservation in all catalytic residues is indicated as follows: the red triangle indicates conserved histidine residues; the purple triangle indicates conserved asparagine and phenylalanine residues as part of the ATP binding box, along with the G1 and G2 boxes. For comparison, the EnvZ catalytic domain (PDB 4PK4) is shown. The highlighted residues are the same as those in the WebLogo and rainbow-colored dimers. (**C**) The predicted topology or histidine kinases are shown per group. The orange bar represents the cell membrane. Accession numbers are indicated. Each image was obtained from Protter [22], and topology was confirmed with TOPCONS [23].

**Figure 3 molecules-28-05042-f003:**
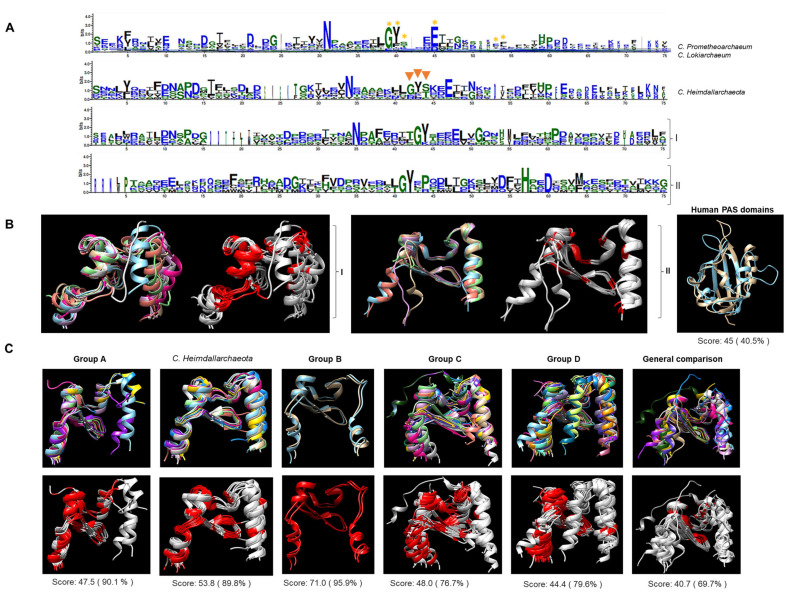
PAS domains in Asgardian TCS kinases are closer to PAS domains from bacterial proteins than eukaryotic proteins. Representative PAS domains used as references were as follows. Aer (P50466), ArcB (P22763), AtoS (Q06067), Bat (Q9HPU8.1), FixL (S39984), StyS (AJ000330), and TodS (U72354) for bacterial examples, and ARNT (AF016053), PER (two PAS domains, P07663), SIM (P05709), and white collar 2 (WC2, Y09119)56. (**A**) Multiple-sequence alignment and WebLogo representation of all PAS domains from *C. Prometheoarchaeum* and *C. Lokiarchaeum*. A yellow asterisk indicates conserved resides in bacterial examples. For *C. Heimdallarchaeota*, orange triangles indicate more diverse regions than *C. Prometheoarchaeum* and *C. Lokiarchaeum* PAS domains. (I) shows the alignment of the bacterial examples used here, and (II) shows the alignment of the eukaryotic examples used here. (**B**) AlphaFold2 models of individual PAS domains from bacteria (I) and eukaryotes (II) were generated and compared. The images show in colors each PAS domain and in light gray and red the conserved residues between each example used (in red). For further comparison, PAS domains from human kinases (PDBs: 1LL8 (light blue) and 1P97 (light brown)) were used. (**C**) individually modeled PAS domains are indicated in the top figure with the alignment shown in colors for each protein per group, and the bottom figure shows the conserved residues in each domain (in red). The last set compares the first two proteins of each group, including the first two of bacterial and eukaryotic origin. Below each alignment, the SALING score is indicated. All models are positioned from **left** (N-terminal end) to **right** (C-terminal end).

**Figure 4 molecules-28-05042-f004:**
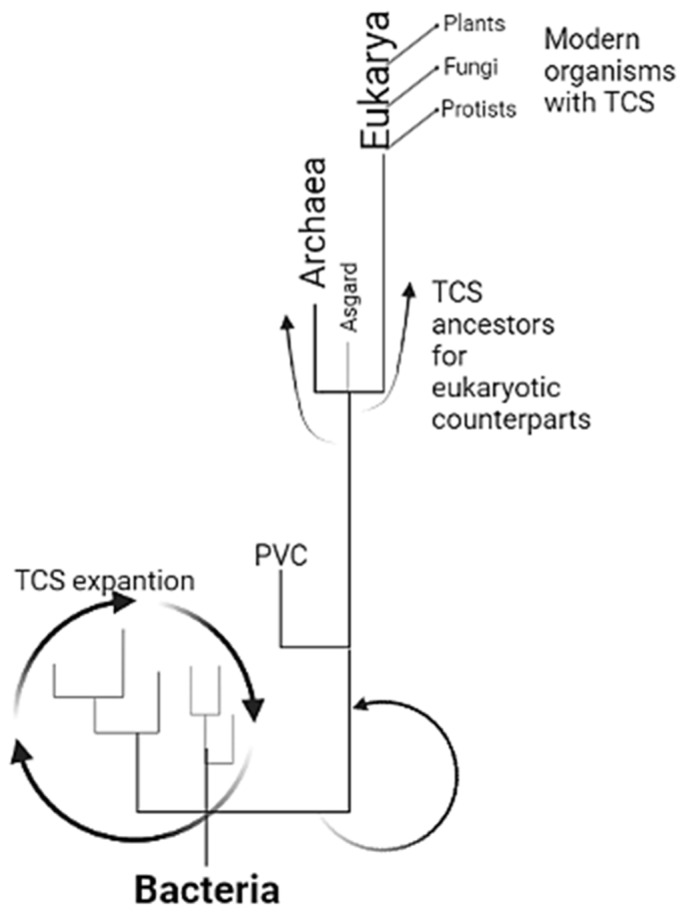
The theoretical model is presented here for expanding TCSs from bacteria to eukaryotes. TCS is the paradigm of signal transduction in bacteria, diversified and vastly expanded in all groups. In the diversification and specialization with environmental restrictions imposed when photosynthetic organisms modified the atmosphere, many organisms relied on anoxic environments, where PVC bacteria may have been enriched and began the specialization of TCSs. This group selected sensor kinases for those with special characteristics that led to overspecialization, resulting in the sensor kinases found in Archaea and Asgard. With eukaryogenesis, TCSs from Asgardian archaea were inherited and further diversified in protists, fungi, and plants, suggesting perhaps several individual attempts that finally worked and established the eukaryotic organisms found today. The scheme was created with BioRender.com, accessed on 1 February 2023.

**Table 1 molecules-28-05042-t001:** Asgardian two-component sensor kinases features.

*C. Lokiarchaeum* (Metagenome)				
Accession Number	Molecular Weight	pI	Aliphatic Index	Domains and Features *
KKK45741.1	52,780.54	6.5	105.88	PAS (2); BaeS; AdeS_HK; HK_VicK; PRK15347; HATPase_C (2)
KKK40441.1	42,288.66	5.74	111.19	PAS; BaeS; AdeS_HK; HK_VicK; HATPase_c (2)
KKK44257.1	58,336.03	7.11	99.39	VicK; HK_WalK; PRK11360; HATPase_c (3)
KKK46537.1	37,305.37	7.02	115.39	BaeS; AdeS_HK; HK_WalK
KKK46579.1	58,196.20	5.1	97.24	Heme pocket; PAS (4); BaeS; HATPase_c (3); AdeS_HK; HK_WalK
KKK43125.1	160,358.88	6.61	98.12	Heme pocket (4); REC_NtrC1-like; ActR_PrrA_rreg; AtoC; PAS (13); GAF_2; BaeS; AdeS_HK, HK_VicK
KKK45609.1	59,864.19	5.17	96.48	PAS_3; BaeS; HK_walK; AdeS_HK:
KKK46428.1	102,728.72	8.59	99.83	BaeS; HK_WalK; Sensor Box 1 (3); PAS (10); AdeS_HK; Heme pocket (3)
KKK46295.1	101,567.75	5.79	99.66	Heme pocket (2); HATPase_EvgS-ArcB-TorS; HK_WalK; PAS (9); PAS_3 (2); AdeS_HK
KKK44457.1	57,183.86	5.35	102.91	Heme pocket; HK_WalK; HK_VicK; HATPase_EvgS-ArcB-TorS; PAS (4)
				Sensory Box 1
KKK40347.1	39,337.96	8.89	117.84	BaeS; BaeS_SmeS; PRK10364; HATPase; HK_WalK; HK _VicK; AdeS_HK
KKK46010.1	34,171.73	7.64	104.08	HisKA; REC; CheY; PRK11100; PRK10365; KdpD
KKK46009.1	131,820.89	5.49	99.11	Heme Pocket (3); 9 PAS/PAC domains in 5 superfamilies; BaeS; AdeS_HK; HATPase
KKK41094.1	112,435.23	5.64	90.78	Putative Active Site (3); Sensory Box (3); Heme Pocket (3); ATP Binding Site (1); ATP Lid (1); Mg-Binding Site (1)
				GxG Motif (1); Contiene 7 PAS en 3 superfamilias; BaeS (1); BaeS_Smes (1); HK-WalK (1); HK-VicK (1); AdeS_HK (1); HATPase (1); HATPase_c (2); PRK09303 (1)
KKK43837.1	90,199.53	6.14	104.49	Putative Active Site (1); Sensory Box (1); Heme Pocket (1); ATP Binding Site (1); ATP Lid (1); Mg-Binding Site (1); GxG Motif (1); Contiene 3 PAS en 1 superfamilia; BaeS (1); BaeS_Smes (1); HK-WalK (1); HK-VicK (1); AdeS_HK (1); HATPase (1); HATPase_c (2); PRK11360 (1)
KKK46292.1	43,755.07	5.13	106.96	Sensory Box (1); ATP Binding Site (1); ATP Lid (1); Mg-Binding Site (1); GxG Motif (1); PAS (1); BaeS (1); BaeS_Smes (1); HK-WalK (1); HK-VicK (1); AdeS_HK (1); HATPase (1); HATPase_c (2); PRK09303 (1)
KKK40193.1	60,981.80	8.54	108.46	COG4251 (Bacteriophytochrome, light-regulated signal transduction histidine kinase)
***C. Prometheoarchaeum* (Sequenced Genome)**				
QEE16613.1	78,222.81	5.52	99.49	Putative active site (1); Sensory box (1); Heme Pocket (1); ATP Binding Site (1); ATP Lid (1); Mg-Binding Site (1); GxG Motif (1); PAS (4); 4 PAS domains in 2 superfamilies; Periplasmic Binding Protein Type 2 (1); BaeS (1); BaeS_Smes (1); HK-WalK (1); HK-VicK (1); TMAO_TorS (1); AdeS_HK (1); HATPase_Walk_EvgS-ArcB-TorS like (1); HATPase_c (2)
QEE16628.1	113,413.22	8.83	95.55	Putative active site (2); Sensory box (2); Heme Pocket (2); ATP Binding Site (1); ATP Lid (1); Mg-Binding Site (1); GxG Motif (1); 7 PAS domains in 3 superfamilies; BaeS (1); BaeS_Smes (1); HK-WalK (1); HK-VicK (1); TMAO_TorS (1); AdeS_HK (1); HATPase_Walk_EvgS-ArcB-T (1); HATPase_c (2); PRK11107 (1)
WP_147663689.1	84,736.02	6.18	100.38	Putative Dimer Interface (1); Sensory Box (2); ATP Binding Site (1); ATP Lid (1); Mg-Binding Site (1); Metal Binding Site (1); GxG Motif (1); Active site (1); 2 PAS in 2 superfamilies; BaeS (1); BaeS_Smes (1); HK-WalK (1); HK-VicK (1); TMAO_TorS (1); AdeS_HK (1); HATPase_AtoS like (1); HATPase_c (2); REC (4 belonging to 1 CheY, 2 REC and 1 Response_reg;
				PRK11360 (AtoS) (1)
WP_147663673.1	80,618.67	5.49	106.34	Heme Pocket (2); PAS (7); 7 PAS domains in 3 superfamilies; TMAO_TorS; BaeS; PRK11091; AdeS_HK; HATPase
QEE16740.1	80,618.67	5.49	106.34	Heme Pocket (2); 7 PAS domains in 3 superfamilies; BaeS; TMAO_TorS; PRK11091; AdeS_HK; HATPase
WP_147664519.1	89,139.07	6.49	111.80	PAS (1); BaeS; AdeS_HK; HATPase_c (2)
WP_147662496.1	87,008.08	5.22	102.36	PAS (8); BaeS; PRK13557; HATPase_c (2); Heme pocket (2)
WP_147664078.1	76,672.69	5.89	100.02	Heme pocket (2); PAS (3); BaeS; HATPase_c (2); AdeS_HK; HK_WalK
QEE16621.1	122,598.94	8.17	102.3	Heme pocket (2); PAS (9); GAF_2; TMAO_torS; HK_VicK; HK_WalK; AdeS_HK; HATPase_c (2)
QEE16756.1	84,736.02	6.18	100.38	PAS (3); BaeS; PRK11360; HATPase_c (2); CheY; REC (2); ActR_PrrA_rreg; Metal binding site
QEE16389.1	77,609.99	4.94	104.50	PBP2_HisK; HK_WalK; HK_VicK
WP_147661239.1	119,091.40	5.86	90.98	PAS (11); BaeS

* indicates that Conserved Domain Search at NCBI found the domains. Parentheses indicate the number of identified domains of the same type.

## Data Availability

The datasets generated for this study are available on request from the corresponding authors. The PDB codes, sequences, and supplementary figures and files are provided in the Appendix A.

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
