# Peer review of "Two-Component System Sensor Kinases from Asgardian Archaea May Be Witnesses to Eukaryotic Cell Evolution"

_molecules, 2023, doi:10.3390/molecules28135042_

Round 1

Reviewer 1 Report

Interesting work contributing in the discussion about early stages of eucaryotes evolution.

I suggest to not overuse abbreviation if it is used one time in the text the sense is low and only complicates.
Specific comments
line: 89-90 it should be residues?
Line 104 abbreviation explanation TMAO
line 106 shows - I propose: illustrates
line 111 remove: well

Table 1 Please add the legend to this table what the asterisk mean, the numbers in brackets are sometimes present sometimes not

Figure1 there are several colours on this figure that do not correspond to the legend please verify what is indicated on the tree
Line 274 LECA You used this abbreviation one time in the whole text. What was the sense of that?
line 347 sediments28 or sediments [28] verify
line360 Add the species name for clarity.
Line 363 is the date in brackets necessary?
Line 365  Species names in italics. Please verify the whole text there is no consequence
Line 368 ,,Russum and coworkers’’ seem to fit better

Author Response

Reviewer 1

Thank you for the positive insight provided. In the following lines, we respond to all the concerns the reviewer raises.

Interesting work contributing in the discussion about early stages of eucaryotes evolution.

Thank you so much. The finding that TCS in these archaea is relevant and links the evolution from bacteria to eukaryotic organisms, also suggesting that these bacteria have a complex signal perception machinery.

I suggest to not overuse abbreviation if it is used one time in the text the sense is low and only complicates.

Thank you for pointing this out; we have corrected this issue in the manuscript.

Specific comments

line: 89-90 it should be residues?

Thank you for the observation; this should be motifs and residues. This has been corrected in line 100.

Line 104 abbreviation explanation TMAO

We have added the full name of the compound (Trimethylamine N-oxide), now line 115.

line 106 shows - I propose: illustrates

Thank you for the suggestion; this has been corrected in line 118.

line 111 remove: well

Thank you for the observation; this is now in line 123.

Table 1 Please add the legend to this table what the asterisk mean, the numbers in brackets are sometimes present sometimes not

We apologize for this omission, we have added the legend in lines 128 and 129. The legend should have stated the following: *Indicates that Conserved Domain Search at NCBI found the domains. Parenthesis indicates the number of the identified domains of the same type.

Figure1 there are several colours on this figure that do not correspond to the legend please verify what is indicated on the tree

Thank you for pointing this out. We have corrected the figure legend in lines 160-168. We apologize for the lack of clarity in this figure.

Line 274 LECA You used this abbreviation one time in the whole text. What was the sense of that?

Thank you for pointing this out; indeed, we only mentioned LECA in the first line of the introduction and then again in this line. We have modified this to the whole meaning of LECA.

line 347 sediments28 or sediments [28] verify

Thank you for pointing this out, we have corrected to “sediments [28]” (line 378)

line360 Add the species name for clarity.

Thank you so much for pointing this out; this has been corrected in lines 391-392.

Line 363 is the date in brackets necessary?

Thank you for pointing this out; we have deleted the reference date as there is no need for this since we have the reference next to it. We have change this and the Davos reference that had the same issue.

Line 365  Species names in italics. Please verify the whole text there is no consequence

The complete text has been corrected, there were also other missing italics. We apologize for this issue.

Line 368 ,,Russum and coworkers’’ seem to fit better

Thank you for the observation; this has been corrected.

Reviewer 2 Report

Overall, this research article represents an interesting investigation on “Two-component system sensor kinases from Asgardian archaea may be witnesses to eukaryotic cell evolution.” The article provides interesting findings about the analysis of the sequence and predicted structural features of two-component system (TCS) sensor kinases in Asgardian archaea. The authors highlight the importance of studying prokaryotic features in Asgardian archaea, particularly in relation to their potential role in the origin of eukaryotic lifeforms. Overall, the manuscript is well-written and provides valuable insights into the evolutionary relationship between Asgardian archaea and bacteria. However, there are some concerns and suggestions for improvement that need to be addressed before the manuscript can be considered for publication.

Abstract is logical, providing a concise summary of the findings. The introduction provides sufficient background on Asgardian archaea and their potential links to eukaryotic lifeforms. However, it would be helpful to clearly state the specific objectives of this study. What specific research questions were addressed? This will help readers better understand the motivation behind the analysis and the significance of the findings.

The results section needs to be revised, the results seem more like discussion, better to add subheadings in results rather than presenting all results in one heading. It makes manuscript hard to understand for readers. The results section would benefit from clearer organization and the inclusion of subheadings. It is important to present the results in a structured and easily understandable manner. Divide the results section into logical subheadings based on the different aspects of the analysis or the specific research questions addressed.

While the findings presented in the manuscript are intriguing, they would benefit from additional contextualization. How do these findings contribute to our understanding of the evolutionary relationship between Asgardian archaea and bacteria? What are the potential implications of the identified TCS kinases in the transition from bacteria to eukaryotic organisms? The authors should expand on the significance and broader implications of their results to provide a clearer link between their findings and the overarching research topic.

The methods section lacks sufficient detail for readers to fully understand the experimental procedures and data analysis techniques employed in this study. Please provide more information on how the metagenome assemblies and genomic assembly were obtained. Additionally, describe the criteria used for selecting the TCS sensor kinases for analysis and provide a rationale for their inclusion. This will enhance the reproducibility of the study and allow readers to evaluate the robustness of the findings.

The authors should acknowledge the limitations of their study and discuss potential avenues for future research. For example, are there any caveats associated with the methodology used? Are there other aspects of Asgardian archaea that warrant further investigation? By addressing these points, the authors can demonstrate a comprehensive understanding of the field and highlight the potential impact of their work on future studies.

Several typographical and grammatical errors were observed throughout the manuscript. A careful proofreading is necessary to correct these errors and enhance the readability of the manuscript.

Overall, the manuscript presents a valuable study investigating TCS sensor kinases in Asgardian archaea and their potential evolutionary significance. Addressing the above-mentioned concerns and suggestions would significantly improve the quality and impact of the manuscript. I recommend revisions before considering it for publication. With some minor improvements, this study could be even more impactful and informative for researchers in this area.

Several typographical and grammatical errors were observed throughout the manuscript. A careful proofreading is necessary to correct these errors and enhance the readability of the manuscript.

Author Response

Reviewer 2

Thank you for the positive insight provided. In the following lines, we respond to all the concerns the reviewer raises.

Overall, this research article represents an interesting investigation on “Two-component system sensor kinases from Asgardian archaea may be witnesses to eukaryotic cell evolution.” The article provides interesting findings about the analysis of the sequence and predicted structural features of two-component system (TCS) sensor kinases in Asgardian archaea. The authors highlight the importance of studying prokaryotic features in Asgardian archaea, particularly in relation to their potential role in the origin of eukaryotic lifeforms. Overall, the manuscript is well-written and provides valuable insights into the evolutionary relationship between Asgardian archaea and bacteria. However, there are some concerns and suggestions for improvement that need to be addressed before the manuscript can be considered for publication.

Thank you for the positive insight and is so gratifying that the manuscript transmits the message we intended to transmit. 

Abstract is logical, providing a concise summary of the findings. The introduction provides sufficient background on Asgardian archaea and their potential links to eukaryotic lifeforms. However, it would be helpful to clearly state the specific objectives of this study. What specific research questions were addressed? This will help readers better understand the motivation behind the analysis and the significance of the findings.

Thank you so much for the suggestion; we have included the objectives in lines 62-67. We hope they are clear and concise.

The results section needs to be revised, the results seem more like discussion, better to add subheadings in results rather than presenting all results in one heading. It makes manuscript hard to understand for readers. The results section would benefit from clearer organization and the inclusion of subheadings. It is important to present the results in a structured and easily understandable manner. Divide the results section into logical subheadings based on the different aspects of the analysis or the specific research questions addressed.

Thank you so much for the observation. We intended to present the results section with strong literature support to avoid any misinterpretation on our side or over-excitement of the findings presented here. In this regard, we kindly request to maintain strong theoretical support for the conclusions presented here. Regarding the addition of subheadings, we have modified the results section accordingly.

While the findings presented in the manuscript are intriguing, they would benefit from additional contextualization. How do these findings contribute to our understanding of the evolutionary relationship between Asgardian archaea and bacteria? What are the potential implications of the identified TCS kinases in the transition from bacteria to eukaryotic organisms? The authors should expand on the significance and broader implications of their results to provide a clearer link between their findings and the overarching research topic.

Thank you for the comment. After revising the manuscript, we weren’t sure where the reviewer suggested making a stronger contextualization. We have added in the results section (lines 299-307) more context to the implications of our work, and then in the discussion section; we emphasized the potential of this work (lines 431-435 and 448-458). We hope this makes the contextualization better.

The methods section lacks sufficient detail for readers to fully understand the experimental procedures and data analysis techniques employed in this study. Please provide more information on how the metagenome assemblies and genomic assembly were obtained. Additionally, describe the criteria used for selecting the TCS sensor kinases for analysis and provide a rationale for their inclusion. This will enhance the reproducibility of the study and allow readers to evaluate the robustness of the findings.

Thank you for the suggestions. In the original text, we provided the BioProject accession numbers for each genome and metagenome assemblies, which we queried for annotated genes as sensor kinases; this last information was now added (lines 466-467). We did not discriminate any of the annotated sensor kinases; we decided to analyze all the sequences we found in these genomes (lines 465-466). In all instances, the proteins were contained in fully sequenced contigs (this has been added to line 467).

In the remaining sections, we used well-established tools and provided references for each tool used. We did not generate new analytic tools. This was added in the tools that we did not indicate that we used the default settings. We hope this will clarify this issue.

The authors should acknowledge the limitations of their study and discuss potential avenues for future research. For example, are there any caveats associated with the methodology used? Are there other aspects of Asgardian archaea that warrant further investigation? By addressing these points, the authors can demonstrate a comprehensive understanding of the field and highlight the potential impact of their work on future studies.

Thank you for the suggestion; we have added this request in lines 459-471.

Several typographical and grammatical errors were observed throughout the manuscript. A careful proofreading is necessary to correct these errors and enhance the readability of the manuscript.

Thank you so much for this observation; we have corrected the manuscript accordingly.

Overall, the manuscript presents a valuable study investigating TCS sensor kinases in Asgardian archaea and their potential evolutionary significance. Addressing the above-mentioned concerns and suggestions would significantly improve the quality and impact of the manuscript. I recommend revisions before considering it for publication. With some minor improvements, this study could be even more impactful and informative for researchers in this area.

Thank you so much for the positive comment. We have made all the suggestions indicated by the reviewer. We appreciate the time and thorough revision of the manuscript to improve its contents.
